# Characterization and Functional Analysis of a New Calcium/Calmodulin-Dependent Protein Kinase (CaMK1) in the Citrus Pathogenic Fungus *Penicillium italicum*

**DOI:** 10.3390/jof8070667

**Published:** 2022-06-25

**Authors:** Guoqi Li, Shaoting Liu, Lijuan Wu, Xiao Wang, Rongrong Cuan, Yongliang Zheng, Deli Liu, Yongze Yuan

**Affiliations:** 1Hubei Key Laboratory of Genetic Regulation and Integrative Biology, School of Life Sciences, Central China Normal University, Wuhan 430079, China; lgq@mails.ccnu.edu.cn (G.L.); wuj1214@mails.ccnu.edu.cn (L.W.); wangxiao5429@mails.ccnu.edu.cn (X.W.); crr@mails.ccnu.edu.cn (R.C.); ldl@mail.ccnu.edu.cn (D.L.); 2School of Public Administration, Central China Normal University, Wuhan 430079, China; liusting@mails.ccnu.edu.cn; 3Hubei Key Laboratory of Economic Forest Germplasm Improvement and Resources Comprehensive Utilization, Hubei Collaborative Innovation Center for the Characteristic Resources Exploitation of Dabie Mountains, College of Biology and Agricultural Resources, Huanggang Normal University, Huanggang 438000, China; ylzheng@hgnu.edu.cn

**Keywords:** *P. italicum*, calcium/calmodulin-dependent protein kinase (CaMK), conidiation, virulence, stress tolerance, transcriptome

## Abstract

Calcium (Ca^2+^)/calmodulin-dependent protein kinases (CaMKs) act as a class of crucial elements in Ca^2+^-signal transduction pathways that regulate fungal growth, sporulation, virulence, and environmental stress tolerance. However, little is known about the function of such protein kinase in phytopathogenic *Penicillium* species. In the present study, a new CaMK gene from the citrus pathogenic fungus *P. italicum*, designated *PiCaMK1*, was cloned and functionally characterized by gene knockout and transcriptome analysis. The open reading frame of *PiCaMK1* is 1209 bp in full length, which encodes 402 amino acid residues (putative molecular weight ~45.2 KD) with the highest homologous (~96.3%) to the *P. expansum* CaMK. The knockout mutant Δ*PiCaMK1* showed a significant reduction in vegetative growth, conidiation, and virulence (i.e., to induce blue mold decay on citrus fruit). Δ*PiCaMK1* was less sensitive to NaCl- or KCl-induced salinity stress and less resistant to mannitol-induced osmotic stress, indicating the functional involvement of *PiCaMK1* in such environmental stress tolerance. In contrast, the *PiCaMK1*-complemented strain Δ*PiCaMK1*COM can restore all the defective phenotypes. Transcriptome analysis revealed that knockout of *PiCaMK1* down-regulated expression of the genes involved in DNA replication and repair, cell cycle, meiosis, pyrimidine and purine metabolisms, and MAPK signaling pathway. Our results suggested the critical role of *PiCaMK1* in regulating multiple physical and cellular processes of citrus postharvest pathogen *P. italicum*, including growth, conidiation, virulence, and environmental stress tolerance.

## 1. Introduction

Postharvest citrus are prone to be infected by pathogenic fungi *Penicillium digitatum* and *P. italicum* that cause green mold disease and blue mold disease, respectively. The latter disease induced by *P. italicum* pathogens usually presents higher tolerance to environmental stress conditions, including cold and salinity [1,2,3]. Specifically, *P. italicum* pathogens with faster mycelium growth are undesirably easy to spread and contaminate healthy citrus fruits even under cold-storage conditions [4] and create more severe virulence [5]. Many studies focused on the molecular mechanisms underlying *P. digitatum* growth; sporulation and virulence, including transcription factors [6,7,8,9,10,11]; signaling responses [6,10,12]; cell cycle regulation [13,14]; and environmental adaptations [12,15,16,17,18,19]. However, such regulation mechanisms are rarely elucidated in *P. italicum* species. The latest reports revealed some key factors to control *P. italicum* virulence [20,21]. Nevertheless, more studies are necessary to identify more effectors to regulate *P. italicum* infection.

Calcium ion (Ca^2+^) serves as an essential signal in fungi to regulate many intracellular processes, including hyphal growth, sporulation, cell cycle, nuclear division, pathogenicity (virulence), and stress resistance [22,23,24]. Ca^2+^/calmodulin-dependent protein kinases (CaMKs), a class of Ser/Thr protein kinases, mediate Ca^2+^ signals to modulate diverse biological behaviors. CaMKs have been functionally characterized in mammalian systems and several fungi systems, including *S. cerevisiae*, *A. nidulans*, *S. pombe*, *Colletotrichum gloeosporioides*, *Sporothrix schenckii*, and *N. crassa*. Knockout of *CaMK* isofroms (*CaMK1* and/or *CaMK2*) in yeast *S. cerevisiae* significantly inhibited spore germination and thermo tolerance [25]. *A. nidulans* CaMKs, known as CMKA (i.e., the homolog of *S. cerevisiae* CaMKs), CMKB, and CMKC, are effectors to control the fungal growth, cell cycle (e.g., G1-G2 transition), and nuclear division [26,27]. *S. pombe CaMK1* also contributed to the fungal cell cycle progression [28]. CoPK12, a novel CaMK in the basidiomycetous fungus *Coprinus cinereus*, was required for active mycelial growth [29]. CaMK or CaMK-like protein kinase has been elucidated to be required for full virulence of various pathogenic fungi, including rice blast fungus *Magnaporthe oryzae* [30], *Puccinia striiformis* f. sp. *tritici* (*Pst*) [31], and nematode-trapping fungus *Arthrobotrys oligospora* [24]. Besides virulence regulation, the involvements of CaMKs in fungal responses to environmental stresses have been intensively studied, including oxidative stress response in *Candida albicans* [32], reactive oxygen stress in the *Pst* fungi [31], heat shock and ultraviolet-radiation stresses in *A. oligospora* [24], and low-pH and osmotic stresses in *Candida glabrata* [33]. To date, the role(s) of CaMK(s) in the regulation of growth, sporulation, virulence, and stress tolerance in *Penicillium* pathogens, including *P. italicum* are still unclear and need more studies.

Fungal growth, sporulation, virulence, and stress tolerance can be regulated through Ca^2+^-involved cross-linking mechanisms, including DNA replication and damage repair, cell cycle and nuclear division, and MAPK signaling. *Candida albicans* pathogenicity was correlated to the DNA damage response pathway that also regulated filamentous growth and hyphal formation through cell cycle re-scheming [34]. Deletion of particular DNA polymerase in the pathogenic yeast *Candida albicans* reduced its filamentation and resulted in a virulence change [35]. Morphogenetic cell-fate decisions to develop *Candida albicans* virulence and stress tolerance usually started with chromatin-associated DNA-replication, which was controlled by complex signaling networks including MAPK and Ca^2+^ signaling [36]. These multilayer regulations were reported only in human fungal pathogens, and how they work in plant-pathogenic fungi as a citrus pathogen, *P. italicum* remains to be elucidated.

In the present study, we, for the first time, characterized a *P. italicum* Ca^2+^/calmodulin-dependent protein kinase (*PiCaMK1*) with high homolog to the model yeast CaMKs and identified its functions in growth, sporulation, virulence, and stress tolerance of the citrus pathogenic fungi (i.e., the blue mold pathogen). We also applied Illumina RNA-sequencing to compare wild-type (control) and *PiCaMK1*-defective strains to confirm global regulatory mechanisms underlying the *PiCaMK1* regulation.

## 2. Materials and Methods

### 2.1. Strains and Cultivation Conditions

*P. italicum* strain YN1 defective in gene *ku70* was used as the control in this work. The fungal strain YN1 was highly resistant to DMI-fungicide prochloraz with an EC_50_ value of approximately 30 mg·L^−1^, as previously reported [37]. This prochloraz-resistant *P. italicum* strain was applied as a recipient in *Agrobacterium tumefaciens*-mediated transformation to knock out the target gene (*PiCaMK1*). Fungal strains were cultivated on potato dextrose agar (PDA) or in potato dextrose broth (PDB) at 28 °C for 5 to 7 days, as previously described [38]. The fungal mycelia grown in the liquid PDB at 28 °C for 2 days were collected to prepare genomic DNA and total RNA. Fungal conidia produced on PDA were collected, and after ddH_2_O washing and ~8000× *g* centrifugation, the conidia were re-suspended to equivalent concentrations (~10^7^ spores mL^−1^) for further phenotype analysis. Conidia were also incubated in a sporulation medium to prepare YN1 protoplasts for the transformation of suitable knockout fragments, according to previous protocols [37]. The *A. tumefaciens* strain AGL-1, stored in 30% (*v*/*v*) glycerol at −70 °C, was exploited as a mediator in the fungal transformation to construct a gene-complemented strain. *E. coli* strain DH5α competent cells to carry the pMD18-T vector for gene cloning were commercially purchased (TaKaRa, Dalian, China) and cultivated in Luria Broth (LB) media containing Ampicillin antibiotics, according to the manual instructions. 

### 2.2. Gene Cloning and Sequence Analysis

Based on the unigene sequence of *PiCaMK1* (PITC_025800) in the previous YN1 RNA-seq report [37], a pair of primers, designated as *PiCaMK1*-F and *PiCaMK1*-R (Appendix A), were used to PCR amplify the full coding region of *PiCaMK1* from YN1 genomic DNA and RT-PCR amplify the corresponding open reading frame (ORF) from cDNA template produced by YN1 total RNA extract. The PCR products were ligated with the pMD18-T vector (TaKaRa, Dalian, China), and the plasmid harboring the *PiCaMK1* genomic gene or ORF was subjected to sequencing. SMART software was applied to predict the conserved domains in the PiCaMK1 primary structure. Local BLAST was online processed on the National Center for Biotechnology Information (NCBI) website to search fungal CaMK genes homologous to the present *PiCaMK1*. According to the outputs, multiple sequence alignments were performed using software ClustalX (version 2.0) with selected fungal CaMK protein sequences (Appendix A). Moreover, based on the classical fungal CaMK sequences (Appendix A), the software MEGA (version 7.0) was processed to build a phylogenetic tree using the neighbor-joining method with 1000 bootstrap replicates.

### 2.3. Gene Knockout and Complementation of PiCaMK1

The PiCaMK1 gene was knockout in the YN1 strain using protoplasts-mediated transformation. According to the homologous recombination-based gene-knockout strategy, the DNA fragment for *PiCaMK1* knockout was composed of 5′ and 3′ flanking sequences of gene *PiCaMK1*, i.e., left (L) and right (R) homologous arms, respectively, and an integrated hygromycin B (Hyg)-resistance cassette. The 5′ and 3′ flanking sequences of *PiCaMK1*, ~1.3 kb and ~1.2 kb in size, respectively, were amplified by specific primer pairs (Appendix A). Overlap PCR was applied to integrate these two homologous arms, using the primers *PiCaMK1*-L-F and *PiCaMK1*-R-R (Appendix A). The generated fragment (i.e., L-R) was inserted into the pMD18-T vector for sequencing confirmation. *Hyg*-resistance cassette was amplified by the primers *Hyg*-F and *Hyg*-R (Appendix A). These two primers were both incorporated with recognition sites of two restriction enzymes, *Spe*I and *Nhe*I, facilitating the following insertion to the L-R fragment. The *Hyg*-resistance cassette, PCR-amplified and digested with *Spe*I and *Nhe*I, was ligated with the vector pMD-L-R, which was also digested with those two restriction enzymes, to generate plasmid pMD-L-Hyg-R containing *PiCaMK1*-knockout fragment L-Hyg-R. The recombinant fragment for *PiCaMK1* knockout, verified by DNA sequencing, was PCR-amplified and purified with a final concentration of ~ 1.0 mg·mL^−1^ for the following polyethylene glycol (PEG)-mediated transformation into the control strain (YN1 defective in gene *ku70*) protoplasts. The transformants null in *PiCaMK1* were selected on PDA media with resistance marker Hyg at 50 µg·mL^−1^ (final concentration) and conformed by PCR using specific primers (Appendix A). The *PiCaMK1*-knockout transformants were also confirmed by Southern blot using digoxigenin (DIG)-labeled probe. The probe was PCR-amplified with primers (Appendix A), and the experimental procedure was according to the description by Wu et al. [38].

PiCaMK1 was genetically complemented to the Δ*PiCaMK1* genome through *A. tumefaciens*-mediated fungal transformation. A DNA fragment containing the entire *PiCaMK1* coding region and its corresponding promoter and terminator sequences was PCR amplified using the primer pair *PiCaMK1*-Com-F/*PiCaMK1*-Com-R (Appendix A). The amplified fragment was digested by *Spe*I and *Xho*I and then inserted into the pPK2-Sur plasmid conferring resistance to chlorimuron ethyl. The resulting plasmid pPK2-Sur-PiCaMK1 was transformed into *A. tumefaciens* strain AGL1. Under the transformant AGL1 and Δ*PiCaMK1* co-cultivation, the putative complemented fungal strains (Δ*PiCaMK1*COM) were selected by Sur resistance (i.e., 10 µg·mL^−1^ chlorimuron ethyl) and confirmed by spore PCR and Southern blot with corresponding primer pairs (Appendix A).

### 2.4. Vegetative Growth, Conidiation, and Virulence Experiments

Vegetative growth experiments were conducted on PDA plates with 10 µL conidia suspensions from the control, Δ*PiCaMK1* and Δ*PiCaMK1*COM strains, respectively. For each strain, the 10 µL conidia sample with a final concentration of ~5 × 10^6^ conidia·mL^−1^ was deposited in the center of the PDA plate, and the colony diameter was measured every day for one week. Conidiation capacity was evaluated with 6-day-old cultures grown on PDA plates, as previously described [38]. For each strain, 150 µL of conidia suspension (~1.0 × 10^7^ conidia·mL^−1^) was evenly spread on PDA plates and cultured for 6 days at 28 °C. The culture plugs in a 7 mm diameter were randomly cut and immersed into 1.0 mL of 0.01% (*v*/*v*) Tween 80. After removing hyphal debris, the conidial yield for each strain was microscopically counted in a hemocytometer, and the result was expressed as conidial number per cm^−2^ colony. Virulence assays were conducted on postharvest mandarin orange fruits inoculated with the control, Δ*PiCaMK1* and Δ*PiCaMK1*COM, respectively. Wounds (~3 mm deep) were created with a sterile needle on each fruit peel, and each wound spot was inoculated with 10 μL conidial suspension (~1.0 × 10^7^ conidia·mL^−1^). The blue mold-infected citrus were incubated at 28 °C for 6 days, and the lesion size was determined at 6 days post-inoculation (dpi). All the experiments were performed in triplicate with statistical analysis using Duncan’s range test in SPSS software (version 20.0).

### 2.5. Abiotic Stress Experiments

In order to investigate fungal tolerance to multiple abiotic stresses, 10 μL conidial suspension (~1.0 × 10^7^ conidia·mL^−1^) for each strain was spotted on PDA alone or supplemented with different abiotic-stress reagents, including chemical fungicides (prochloraz and imazalil), NaCl, KCl, D-mannitol, and H_2_O_2_. These fungal cultures, i.e., those from the control, Δ*PiCaMK1,* and Δ*PiCaMK1*COM, were grown on PDA plates at 28 °C for 6 days, and the diameters of fungal colonies were measured at 6 dpi. In the fungicide experiments, the EC_50_ values were calculated based on the colony diameters at gradient fungicide concentrations, as described by Zhang et al. [37]. The gradient concentrations for fungicide prochloraz were 0, 10, 30, 50, and 70 mg·L^−1^, and for imazalil were 0, 5, 10, 15, and 20 mg·L^−1^. The gradient concentrations for NaCl, KCl, and D-mannitol were 0, 0.3, 0.6, 0.9, and 1.2 mol·L^−1^, and for H_2_O_2_ were 0, 2, 4, 6, and 8 mmol·L^−1^. The relative growth in the stress experiments for each strain was determined as the ratio of colony diameter at a given stress concentration relative to that at 0 concentration. All the experiments were performed in triplicate with statistical analysis using Duncan’s range test in SPSS software (version 20.0).

### 2.6. Analysis of PiCaMK1-Mediated Transcriptomes 

The control and Δ*PiCaMK1* mutant strains were grown on PDA plates for 6 days to collect conidial suspension with final concentration 1.0 × 10^7^ conidia·mL^−1^. Then, 200 μL conidial suspension was further cultured in 200 mL PDB for 2 days at 28 °C, and the resulting mycelia of the indicated strains were used for total RNA extraction according to the previous description [37]. The RNA samples, after quality guarantee in integrity and purity, were applied to construct cDNA libraries for sequencing on the Illumina HiSeq X platform (BioMarker Technologies Company Limited, Beijing, China). The resulting clean reads were mapped to *P. italicum* PHI-1 reference genome (GenBank accession number: JQGA01000000) using the software TopHat (version 2.0.11) [39,40]. The clean reads were finally assembled to unigenes through alignment analysis using software Bowtie2 (version 2.2.5) [41]. Unigene function was annotated by homolog analysis in public databases at BLAST E-value ≤ 1 × 10^−5^ and HMMER E-value ≤ 1 × 10^−10^. Software HTSeq (version 0.6.1) was applied to estimate unigene or transcript abundance based on FPKM analysis. Software Plotly (Montreal, Quebec, QC, USA) was processed for heatmap analysis of hierarchically clustered unigenes with Venn diagram-based visualization. Software package DEGSeq R (version 1.12.0) was applied to identify differentially expressed genes (DEGs) between the control and Δ*PiCaMK1* libraries according to the cut-off value |log2 Fold Change| ≥ 1 (*p*-value ≤ 0.005). Down-regulated DEGs were functionally enriched and classified by KOBAS software at the Kyoto Encyclopedia of Genes and Genomes (KEGG) database (http://www.genome.jp/kegg/, accessed on 30 October 2021).

### 2.7. Real-Time Quantitative PCR (RT-qPCR)

In order to validate the gene expression profile in the present transcriptome, forty DEGs were selected and subjected to RT-qPCR analysis. Total RNA was, respectively, extracted from the control and Δ*PiCaMK1* mutant mycelia samples using TRIzol reagent (Thermo, Waltham, MA, USA) and translated into cDNA using PrimeScript^TM^ RT reagent Kit (TaKaRa, Dalian, China), according to the previous method [37]. RT-qPCR was operated in the BIO-RAD CFX96 qPCR system (BioRad, Hercules, CA, USA) with SYBR Green I fluorescent dye detection, cDNA templates (10× dilution), and primer pairs (Appendix A). The relative transcript abundance of the target gene in Δ*PiCaMK1* over that in the control was calculated using the 2^−ΔΔCt^ method [42] with the gene *β-actin* as an internal reference. The experiments were performed in triplicate with three technical repeats, and the results were expressed as relative transcript abundance with mean ± SD (standard deviation). One-way ANOVA and the least significant difference (LSD) test were applied to statistics analysis at * *p* < 0.05 and ** *p* < 0.01.

## 3. Results

### 3.1. Cloning and Sequence Analysis of PiCaMK1 Gene

According to one fungal *CaMK* coding region (ID: PITC_025800) in transcriptome unigene library, referred to as *PiCaMK1* in the present study, the full-length sequence of the kinase gene and its corresponding ORF was PCR-amplified from genomic DNA and corresponding cDNA of *P. italicum* strain YN1, respectively. Sequence analysis indicated that the PiCaMK1 gene had a 1379-bp coding region with three introns (Appendix A). The three introns were located in positions 45–108, 321–374, and 531–582 bp, with sizes 64, 54, and 52 bp, respectively. The PiCaMK1 gene contained a 1209-bp ORF (Appendix A), encoding a putative protein of 402 amino acids that shared the highest sequence identity with *P. expansum* CaMK (Accession no.: XP_016598664) (Figure 1). Multiple sequence alignments displayed 11 consensus domains in the PiCaMK1 and the other fungal CaMKs (Figure 1), with a highly conserved sequence profile for each domain. Based on amino acid alignments among the selected fungal CaMKs (Appendix A), a phylogenetic tree was constructed, revealing the closest relationship between PiCaMK1 and its ortholog from *P. expansum* (Figure 2).

### 3.2. Knockout of PiCaMK1 Gene and Complementation in P. italicum

The PiCaMK1 gene was replaced by a Hyg-resistance cassette via homologous recombination (Figure 3A). *PiCaMK1*-knockout fragment L-Hyg-R was transformed into the YN1 strain via protoplasts-mediated fungal transformation. Fungal transformants appearing on Hyg-containing PDA were subjected to PCR-based screening for *PiCaMK1*-knockout mutants (Figure 3B), using two primer pairs, *PiCaMK1*-Ko-F/*PiCaMK1*-Ko-R and *PiCaMK1*-Diag-F/*PiCaMK1*-Diag-R, respectively (Appendix A). Two transformants with *PiCaMK1* deletion (i.e., Δ*PiCaMK1*-1^#^ and Δ*PiCaMK1*-2^#^) were available from ~50 transformants after two rounds of PCR screening. There were no significant differences in any phenotype for the two knockout mutants, so one of them was selected as representative in the following experiments. The gene *PiCaMK1* was introduced into the Δ*PiCaMK1* genome using *A. tumefaciens*-mediated fungal transformation (Figure 3A). Transformation of a functional copy of *PiCaMK1* into Δ*PiCaMK1* generated a complementation strain (designated Δ*PiCaMK1*COM) with a wild-type *PiCaMK1* allele by PCR confirmation using the primer pair *PiCaMK1*COM-F/*PiCaMK1*COM-R (Figure 3C). Δ*PiCaMK1*COM with *Sur*-resistance displayed similar phenotypes in growth, conidiation, and virulence, as compared to the control strain (see below for details). The achievement of *PiCaMK1*-knockout (Δ*PiCaMK1*) and -complementation (Δ*PiCaMK1*COM) was further confirmed by Southern blot hybridization (Figure 3D), using PCR-amplified fragment (310 bp) from the target gene as a probe (Appendix A).

### 3.3. PiCaMK1 Is Required for Vegetative Growth and Conidiation

The knockout of *PiCaMK1* reduced fungal vegetative growth by ~52.3%, and such a defective phenotype could be reversed in full by the gene *PiCaMK1* complementation (Figure 4A). For detail, the growth rate of Δ*PiCaMK1* on PDA plates was an average of 2.5 mm, increasing in colony diameter every day for one week. This growth parameter, observed in Δ*PiCaMK1*, was much lower than those of the control and Δ*PiCaMK1*COM strains (i.e., ~6.7 mm colony diameter per day on average) (Figure 4B). These results indicated the requirement of PiCaMK1 in the *P. italicum* vegetative growth. On the other hand, Δ*PiCaMK1* produced ~5.6 × 10^7^ conidia cm^−2^ at 6 dpi, which was much smaller than that of the control (~8.5 × 10^7^ conidia cm^−2^) (Figure 4C). Meanwhile, the conidiation of Δ*PiCaMK1*COM (~8.4 × 10^7^ conidia cm^−2^) was almost fully restored to the control level (Figure 4C). These results indicated the requirement of PiCaMK1 in the *P. italicum* conidiation.

### 3.4. PiCaMK1 Is Required for Full Virulence

The role of the PiCaMK1 gene in fungal virulence was investigated in postharvest orange fruits infected by the control and mutant *P. italicum* strains. At 6 dpi, the significantly larger disease spots (or rotted area) were observed in the control and *PiCaMK1*-complemented strains, as compared to those of the gene-knockout mutants (Figure 5A). The mean diameter of the macerated lesions of the postharvest fruits incubated with the Δ*PiCaMK1* conidial suspensions at 6 dpi was ~41.3 mm; in contrast, the mean diameter was ~61.8 mm for the orange fruits incubated with control conidial suspensions (Figure 5B). By considering the mean diameter as a virulence indicator, the Δ*PiCaMK1* virulence to the postharvest citrus was decreased by ~33%. The complementation of gene *PiCaMK1* can almost totally restore the fungal virulence with a mean diameter of ~ 61.6 mm, comparable to that of the control strains (Figure 5B). These results indicated the requirement of PiCaMK1 for the full virulence of *P. italicum*.

### 3.5. PiCaMK1 Has No Contribution to DMI-Fungicide Resistance

In order to investigate the role of PiCaMK1 in the regulation of DMI-fungicide resistance, the control and mutant *P. italicum* strains were grown on the prochloraz- and imazalil-supplemented PDA plates. The decreasing rate in colony diameter at increasing DMI-fungicide concentrations (i.e., 0~70 mg·L^−1^ for prochloraz and 0~20 mg·L^−1^ for imazalil) was not significantly different for the control, *PiCaMK1*-knockout, and *PiCaMK1*-complemented strains (Figure 6A). The EC_50_ values towards prochloraz were all around 30 mg·L^−1^, with no significant difference in those *P. italicum* strains (Figure 6B). Meanwhile, the EC_50_ values towards imazalil were all around 16 mg·L^−1^, also with no significant difference in those *P. italicum* strains (Figure 6C). Hence, the knockout of *PiCaMK1* had no effect on the *P. italicum* resistance to the two common DMI-fungicides. These results indicated that PiCaMK1 did not contribute to the fungal DMI-fungicide resistance.

### 3.6. The Role of PiCaMK1 in Stress Tolerance of P. italicum

The responses of Δ*PiCaMK1* to different stress conditions, as compared to those of the control and complementation strains, were investigated on the PDA plates with different stress agents, including chloride salts (NaCl and KCl), D-mannitol, and hydrogen peroxide (H_2_O_2_) (Figure 7A). The growth of the control strain at 0~0.3 mol·L^−1^ NaCl, relative to that without NaCl treatment (defined as 100% relative growth), was increased to the maximum (i.e., ~150% relative growth) and then decreased at 0.3~1.2 mol·L^−1^ NaCl (Figure 7B). However, such fluctuation in vegetative growth at increasing NaCl concentrations was not observed in the Δ*PiCaMK1* (Figure 7B), indicating that the *PiCaMK1* deletion remarkably lowered the fungal sensitivity to NaCl stress. A similar effect of *PiCaMK1* deletion was observed in the KCl treatments (Figure 7C). These results indicated the positive role of *PiCaMK1* in the *P. italicum* response to salt-induced salinity stresses.

On the other hand, at mild D-mannitol concentrations (0~0.3 mol·L^−1^), the growth acceleration of Δ*PiCaMK1* by the osmoregulator treatment was well identical to those of the control and the *PiCaMK1*-complemented strains (Figure 7D). However, at higher D-mannitol concentrations, such as at 0.6 and 0.9 mol·L^−1^, the osmotic stimulation of fungal growth was obviously weakened in the Δ*PiCaMK1* as compared to those of the control and *PiCaMK1*-complemented strains (Figure 7D), indicating the positive contribution of PiCaMK1 to the *P. italicum* tolerance to such osmotic regulator of stress.

In contrast, the relative growth curves at the present H_2_O_2_ treatments were very similar for the three *P. italicum* strains (Figure 7E). Thus, the deletion of *PiCaMK1* did not influence the fungal response to the H_2_O_2_ treatment at concentrations ranging from 0 to 8 mmol·L^−1^, as compared to those of the control and *PiCaMK1*-complemented strains, indicating the irrelevance of PiCaMK1 with the *P. italicum* tolerance to the oxidative stress.

### 3.7. Transcriptome Analysis and KEGG Enrichment of DEGs

The RNA samples from the control and Δ*PiCaMK1* conidial suspensions were subjected to transcriptome analysis. Illumina sequencing provided 20,320,426 clean reads for the control sample with Q30 ≥ 95.0% and 23,268,989 clean reads for the Δ*PiCaMK1* sample with Q30 ≥ 95.1%. Based on reference genome PHI-1, the clean reads were finally assembled into ~9100 unigenes for the two *P. italicum* samples. All the unigene expression levels were determined by FPKM values, and based on these values, a hierarchical cluster (i.e., heat map) analysis was performed to visualize DEG profiles between the control and Δ*PiCaMK1* samples (Figure 8A). Using |log_2_(Fold Change)| ≥ 1 and *p*-value ≤ 0.005 as the cut-off values, both the volcano plot and the MA plot analysis identified 364 DEGs in the *PiCaMK1*-deleted strain as compared to the control, including 165 up-regulated and 199 down-regulated (Figure 8B,C). Further, KEGG enrichments classified the 165 up-regulated DEGs into 53 pathways with no significant enrichment, as shown in the top 20 in Figure 9A. In contrast, the 199 down-regulated DEGs were classified by KEGG enrichments into two significantly enriched pathways in the total 53 pathways, i.e., ‘DNA replication’ (ko03030) and ‘Cell cycle-yeast’ (ko04111) (Figure 9B).

As listed in Table 1, the down-regulated DEGs enriched in ‘DNA replication’ were the enzyme and factor-encoding genes responsible for eukaryote DNA biosynthesis, including DNA primase, DNA polymerase, and DNA replication licensing factors. Regarding ‘cell cycle-yeast’, the second significant enrichment of the KEGG pathway, the down-regulated DEGs in the Δ*PiCaMK1* strain were functionally associated with DNA replication, nuclear condensing and division, mitotic spindle regulation, and cell-cycle control (Table 1), including DNA replication licensing factors, Nuclear condensin complex Smc2, condensin complex subunits, mitotic spindle checkpoint protein (Mad2), and cell-cycle checkpoint protein kinase. According to the hierarchical clustering results (Figure 8A), the down-regulated genes in the *PiCaMK1*-deleted strain were also enriched into additional KEGG pathways contributing to the ‘DNA replication’ and ‘cell cycle-yeast’, including purine metabolism (ko00230), pyrimidine metabolism (ko00240), base excision repair (ko03410), nucleotide excision repair (ko03420), mismatch repair (ko03430), meiosis (ko04113), and MAPK signaling pathway (ko04011) (Table 1). On the other hand, Table 1 further shows a list of KEGG-enriched DEGs down-regulated after the gene *PiCaMK1* knockout in the hierarchical clustering with the ‘DNA replication’ and ‘cell cycle-yeast’ pathways, such as cytochrome c oxidase assembly protein in the pathway ‘oxidative phosphorylation’ (ko00190) and aldehyde dehydrogenase in the pathway ‘carotenoid biosynthesis’ (ko00906). In summary, all those down-regulated DEGs identified in the KEGG enrichments were putatively involved in the PiCaMK1 regulation of the fungal phenotypes, including vegetative growth, sporulation, virulence, and environmental stress tolerance.

### 3.8. RT-qPCR Validation of DEGs

The present study selected 40 DEGs from the KEGG enrichment of down-regulated genes to perform RT-qPCR validation. The transcript abundances of all these 40 DEGs were ~50% to 90% decreased in the Δ*PiCaMK1* strain as compared to the control strain (Figure 10). Thus, regarding the expression profile of the selected DEGs, the result of the present RT-qPCR was in agreement with that of the transcriptome analysis.

## 4. Discussion

Multiple CaMK genes have been cloned and identified in the fungi kingdom. Thus far, fungal homologues of the CaMKs have been extensively found in *S. cerevisiae* [25,43,44], *A. nidulans* [26,27,45], *S. pombe* [28], *Colletotrichum gloeosporioides* [46], *Sporothrix schenckii* [47], *N. crassa* [23,48,49], *Puccinia striiformis* f. sp. tritici [31], *Arthrobotrys oligospora* [24], and *Candida glabrata* [33]. Most of the fungal CaMKs, including those from *S. cerevisiae* (*Sc*CMK1 and *Sc*CMK2) and the model filamentous fungus *A. nidulans* (*An*CMKA), exhibit high sequence homology with mammalian CaMKII [23]. In the present study, the *Pi*CaMK1 isolated from the postharvest citrus pathogen *P. italicum* also showed much higher sequence homology (~35%) with mammalian CaMKII than those of the other mammalian CaMKs. Based on the multiple sequence alignments (Figure 1), the amino acid sequence of *Pi*CaMK1 was highly identical to those of *S. cerevisiae* and *A. nidulans* CaMKs. Moreover, all the 11 domains’ consensus in CaMKs were highly conserved in the selected fungal candidates (Figure 1), especially in the N-terminal catalytic domain, autoinhibitory domain, and CaM-binding domain. Thus, the present *Pi*CaMK1 can be classified into the type-II CaMKs, a class of multifunctional kinases with broad substrate specificity [50,51]. Among the available CaMKs from *Penicillium* species, the *Pi*CaMK1 was clustered with *Pe*CaMK in the closest evolution distance, according to the phylogeny analysis (Figure 2). Interestingly, the evolutionary distance between *P. italicum* and *P. digitatum*, the most harmful phytopathogenic fungi of citrus, was far from that between *P. italicum* and *P. expansum*, as shown in the CaMKs phylogenetic tree (Figure 2). Such phylogenetic characteristics might indicate the fruit spectrum of different *Penicillium* species to infect. In summary, the present study provided the first report on the *Pi*CaMK gene sequence, and more *Pi*CaMK gene(s) in the *P. italicum* genome need further investigation.

CaMKs have been extensively characterized to play regulatory roles in various fungal processes, including growth, conidial development, virulence, and stress tolerance. The early studies indicated the requirement of CMKA in the *A. nidulans* hyphal growth and nuclear division [26] and CMK2 in the *S. cerevisiae* spore germination [25]. Similar results were observed in the present work that the Δ*PiCaMK1* showed defective in vegetative growth and conidiation (Figure 4). Such physiological defects have been associated with the arrest of the cell cycle and the prior DNA synthesis by the gene knockout of CaMK(s) [28,48,52]. The present transcriptome analysis and KEGG enrichments suggested simultaneous down-regulation of genes in ‘DNA replication’ and ‘cell cycle’ pathways in the *PiCaMK1*-deleted strain (Table 1 and Figure 9), including the genes encoding DNA primase, DNA polymerases δ and ε, DNA replication licensing factors (i.e., Mcm2, Mcm3 and Mcm6), condensin complex subunit, cell-cycle checkpoint protein kinase, nuclear condensin complex Smc2, and mitotic spindle checkpoint protein Mad2 (Table 1 and Figure 10). Thus, the *PiCaMK1* participated in regulating fungal growth and sporulation at the genetic and cellular levels via those key enzymes and regulatory protein elements.

The present study also reported a significant decrease in the *P. italicum* virulence towards postharvest citrus fruits after the gene *PiCaMK1* knockout (Figure 5). The requirement of CaMK(s) in the full fungal virulence was previously reported in the rust fungi *P. striiformis* f. sp. tritici [31] and the nematode-trapping fungus *A. oligospora* [24]. Virulence mechanisms have been implicated as sophisticated, including DNA replication [35,53], DNA damage responses (e.g., DNA repairs during infective hypha formation) [34,54], and cell-cycle progression regulation [55]. The Δ*PiCaMK1* mutant with reduced virulence also had a much lower transcript abundance of genes involved in ‘DNA replication’, ‘cell cycle’, ‘meiosis’, ‘base excision repair’, ‘nucleotide excision repair’, and ‘mismatch repair’ pathways, according to the results of KEGG enrichment (Table 1) and RT-qPCR validation (Figure 10). In addition, some key enzyme-encoding genes responsible for the biosynthesis of (deoxy)nucleotides were down-regulated in the Δ*PiCaMK1* strain, as shown in the KEGG pathways ‘purine metabolism’ and ‘pyrimidine metabolism’ (Table 1). Among them, ribonucleotide reductase was implicated as essential for pathogen growth and virulence via cell division control [53]. Uracil phosphoribosyltransferase was proposed as a potential virulence factor during *B. cinerea* growth and infection [56]. Xanthine dehydrogenase, with the function in purine salvage synthesis, has been verified to regulate iron homeostasis that contributed to *F. graminearum* virulence [57]. All these enzyme-encoding genes were down-regulated in the *PiCaMK1*-deleted strain, indicating the role of PiCaMK1 as a virulence factor in regulating the other virulence factors. The CaMK-mediated interactions between these virulence factors need to be intensively studied to gain a new mechanism underlying the control of *P. italicum* virulence.

The ca^2+^-calcineurin pathway was proposed to mediate fungal azole resistance [58,59]. However, CaMKs are not located in the Ca^2+^-calcineurin pathway [60]. As a result, the knockout of *PiCaMK1* did not alter the *P. italicum* resistance to azole fungicides (Figure 6), indicating the irrelevance of *PiCaMK1* with fungal azole resistance. Our transcriptome results also provided evidence that the DEGs between the control and Δ*PiCaMK1* strains did not include any fungicide-resistance gene. Sine multiple *PiCaMK* unigenes have been assembled [37], further studies are needed to check the role of other *PiCaMK* in the fungicide-resistance regulation.

The *PiCaMK1*-deleted strain showed much less sensitivity to the NaCl and KCl stresses (Figure 7A–C), as well as lower tolerance to D-mannitol-induced osmotic stress (Figure 7A,D). Yeast protein kinase A and CMK2 served as regulators in response to salt stresses [61,62]. Similar results were obtained in the present Δ*PiCaMK1* mutant, especially at 0.3~0.9 mol·L^−1^ salt conditions. MAPK pathway, as well as energy metabolism, were in close association with the yeast CMK2 regulation [61,62], which can be verified in our transcriptome analysis that the gene encoding phosphatidylinositol 4-kinase in the MAPK pathway and the genes encoding cytochrome c oxidase assembly protein and mitochondrial F_1_/F_0_-ATP synthase in oxidative phosphorylation were down-regulated in the *PiCaMK1*-deleted strain (Table 1). Fungal osmotic regulation is usually mediated by the Ca^2+^-calcineurin signaling pathway [60]. A recent report proposed a correlation of the CgCmk1 with the osmotic regulation of *C. glabrata* under low-pH stress conditions [33]. In the present study, the gene *PiCaMK1* also participated in the D-mannitol-induced osmotic regulation of *P. italicum*, especially at high concentrations (e.g., 0.6~0.9 mol·L^−1^). D-mannitol is contributed to conidial germination and mycelial growth [63,64] and also to fungal pathogenicity [65]. Thus the *PiCaMK1*-mediated mannitol tolerance might be necessary for the kinase to regulate *P. italicum* sporulation and virulence.

In summary, the present study cloned a PiCaMK gene (*PiCaMK1*) from the citrus pathogenic fungus *P. italicum* and classified the *PiCaMK1* into the CaMKII gene family with the closest phylogenetic relationship with that of *P. expansum*. Gene knockout and complementation analysis indicated the requirement of the *PiCaMK1* in the fungal vegetative growth, sporulation, full virulence, and responses to salt (salinity) and mannitol (osmotic) stresses. Transcriptome analysis suggested the involvement of DNA biosynthesis and repair, cell cycle, and some stress-responsive pathways in the *PiCaMK1* regulation.

## Figures and Tables

**Figure 1 jof-08-00667-f001:**
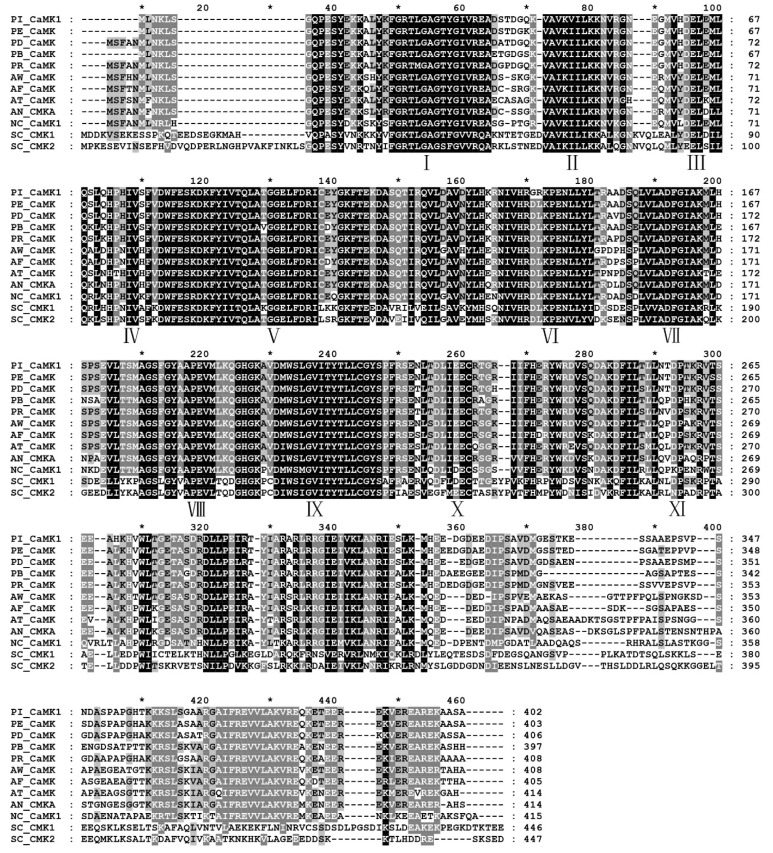
Multiple sequence alignments of fungal calcium/calmodulin-dependent protein kinases (CaMKs). Amino acid sequence of CaMKs from the selected fungi (Appendix A) were compared using the software ClustalX 2.1 and GeneDoc. Conserved amino acid residues are indicated in black (100%), dark gray (>80%), and light gray (>60%). I–XI represents conserved CaMK domains. The asterisk (*) indicates the middle position between two neighboring numbers above the selected sequences.

**Figure 2 jof-08-00667-f002:**
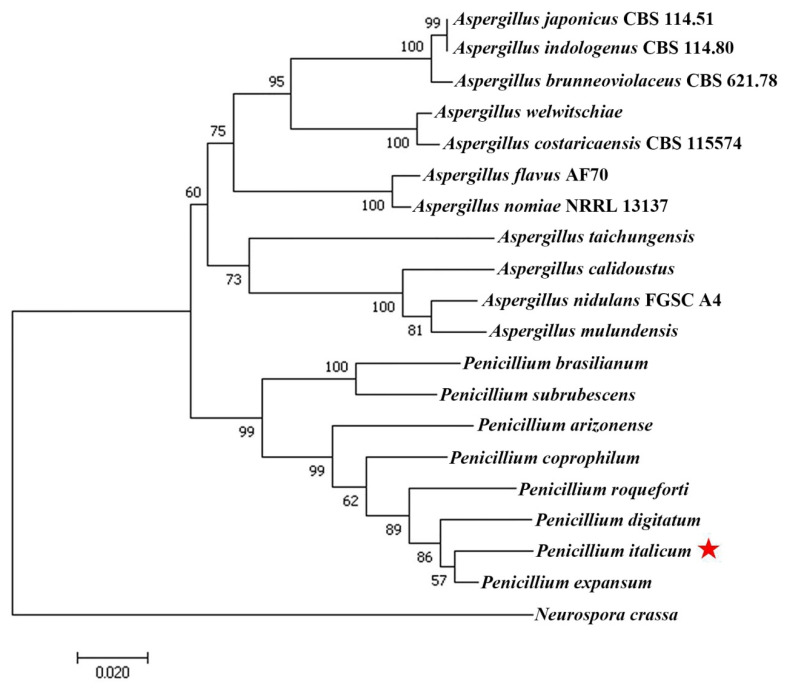
Phylogenetic analysis of PiCaMK1 among fungal CaMKs. Phylogenetic analysis of the CaMKs from the selected fungi (Appendix A) was performed using the minimum evolution method with 500 bootstrap replications in the phylogeny test by MEGA7.0 software. CaMKs are described by Genbank accession number, organism, and phylum. Bars indicate the scale of genetic distances. The red star indicates the position of PiCaMK1 in this study.

**Figure 3 jof-08-00667-f003:**
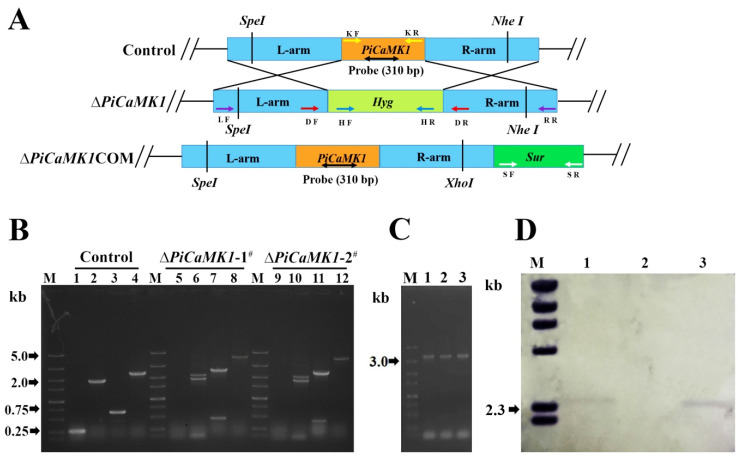
Construction and verification of *PiCaMK1*-knockout and -complementation mutants. (**A**) Schematic diagram to construct the *PiCaMK1*-knockout and -complementation mutants. (**B**) Image of DNA fragments amplified from genomic DNA of the control and the *PiCaMK1*-knockout mutants (Δ*PiCaMK1*). M: DNA marker DS5000; lanes 1, 5, and 9: PCR fragments with primers *PiCaMK1*-Ko-F/R; lanes 2, 6, and 10: PCR fragments with primers Hyg-F/R; lanes 3, 7, and 11: PCR fragments with primers *PiCaMK1*-Diag-F/R; lanes 4, 8, and 12: PCR fragments with primers *PiCaMK1*-L-F and *PiCaMK1*-R-R. (**C**) Image of DNA fragments amplified from genomic DNA of the control and the Δ*PiCaMK1*COM strains. M: DNA marker DS5000; lanes 1, 2, and 3: PCR fragments with primers *PiCaMK1*-COM-F/R. (**D**) Southern blot hybridization of fungal genomic DNA after digestion with *Xba* I. The digested DNA fragments were electrophoresed in an agarose gel, then blotted to a nylon membrane, and finally hybridized to a *PiCaMK1*-specific probe (310 bp in size). M: DIG-labeled DNA marker; lanes 1, 2, and 3: the control, Δ*PiCaMK1*, and Δ*PiCaMK1*COM.

**Figure 4 jof-08-00667-f004:**
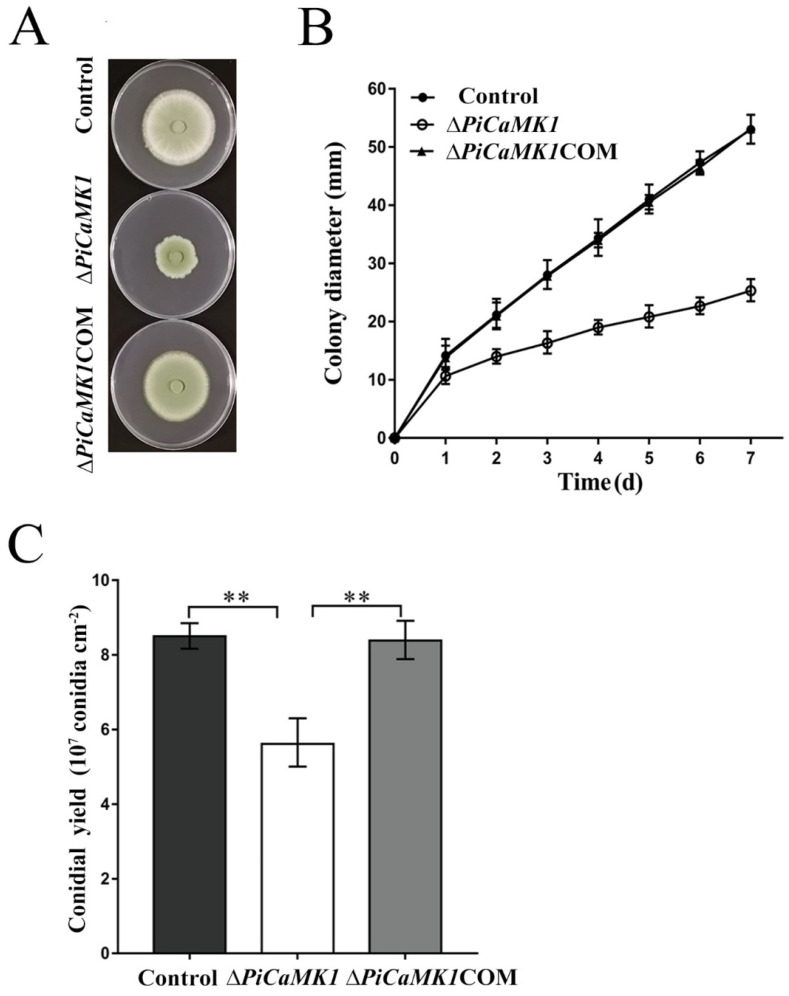
*PiCaMK1* is required for the *P. italicum* vegetative growth and conidiation. (**A**) Images of the control, Δ*PiCaMK1,* and Δ*PiCaMK1*COM strains grown on potato dextrose agar (PDA) for 7 days. (**B**) Vegetative growth rates of the control, Δ*PiCaMK1,* and Δ*PiCaMK1*COM strains grown on PDA. (**C**) Conidia yield quantification of the different *P. italicum* strains grown on PDA for 6 days. The data presented are the mean and standard deviation of three independent experiments with at least three replicates (** *p* < 0.01).

**Figure 5 jof-08-00667-f005:**
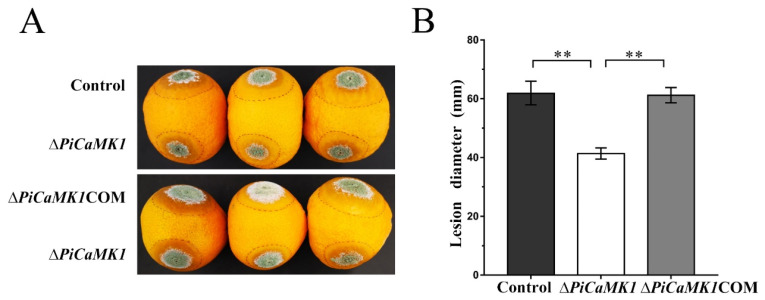
*PiCaMK1* is required for the *P. italicum* full virulence. (**A**) Images of virulence assays on the postharvest citrus fruits infected by the control, Δ*PiCaMK1,* and Δ*PiCaMK1*COM strains. The postharvest citrus fruits were inoculated with 10 µL of conidial suspension (1 × 10^7^ conidia·mL^−1^) from the different *P. italicum* strains, and the lesion size was determined at 6 days post-inoculation (dpi). (**B**) Quantification of the lesion size on the citrus fruits. The data presented are the mean and standard deviation of three independent experiments with at least three replicates (** *p* < 0.01).

**Figure 6 jof-08-00667-f006:**
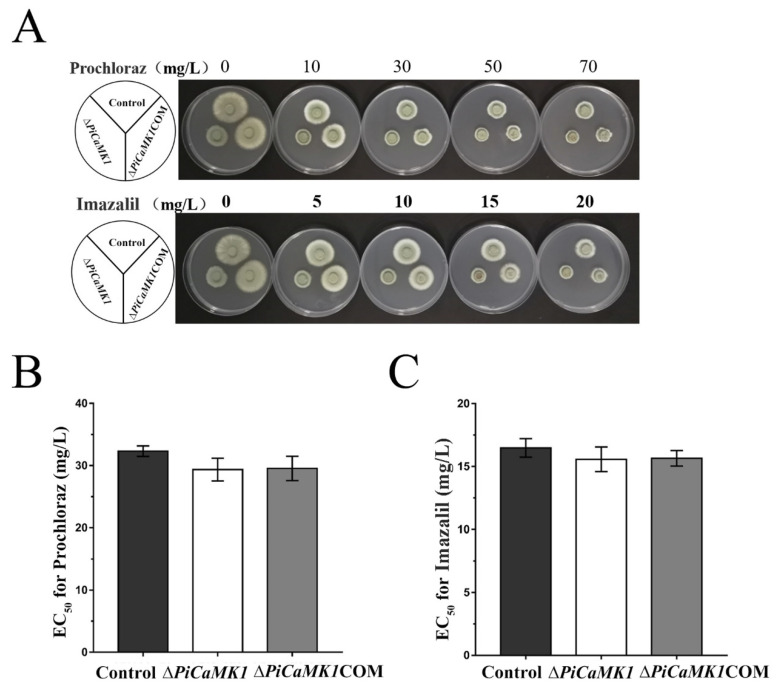
*PiCaMK1* is not required for the *P. italicum* resistance to the DMI fungicides. (**A**) Images of the *P. italicum* growth on the PDA plates with the increasing DMI-fungicide concentrations. DMI fungicides prochloraz and imazalil were used in the experiments. Mycelial plugs from the control, Δ*PiCaMK1,* and Δ*PiCaMK1*COM colonies were cultivated individually on a PDA medium with the indicated concentrations of DMI fungicides, i.e., prochloraz and imazalil, respectively, and the fungal colony diameters were recorded at 6 dpi at 28 °C. (**B**) Prochloraz EC_50_ assays. (**C**) Imazalil EC_50_ assays. The data presented are the mean and standard deviation of three independent experiments with at least three replicates.

**Figure 7 jof-08-00667-f007:**
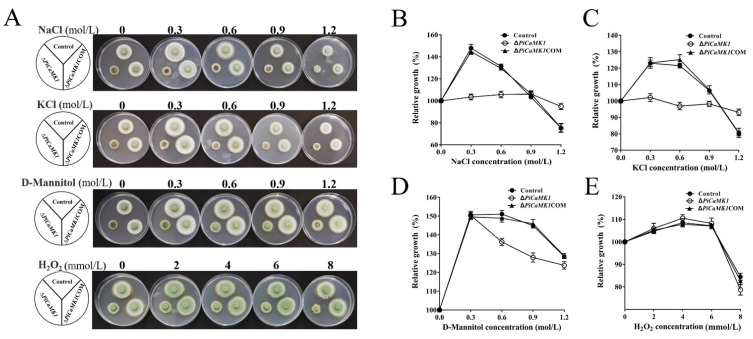
Effects of *PiCaMK1* on stress tolerance of the *P. italicum* to KCl, NaCl, D-mannitol, and H_2_O_2_. (**A**) Images of the *P. italicum* growth on the PDA plates with the increasing concentrations of NaCl, KCl, D-mannitol, and H_2_O_2_, respectively. The mycelial plug operation and cultivation process were as described in Figure 6 legend. (**B**) The effects of NaCl on the relative growth of the different *P. italicum* strains, i.e., the control, Δ*PiCaMK1,* and Δ*PiCaMK1*COM strains. (**C**) The effects of KCl on the relative growth of the different *P. italicum* strains. (**D**) The effects of D-mannitol on the relative growth of the different *P. italicum* strains. (**E**) The effects of H_2_O_2_ on the relative growth of the different *P. italicum* strains. The data presented are the mean and standard deviation of three independent experiments with at least three replicates.

**Figure 8 jof-08-00667-f008:**
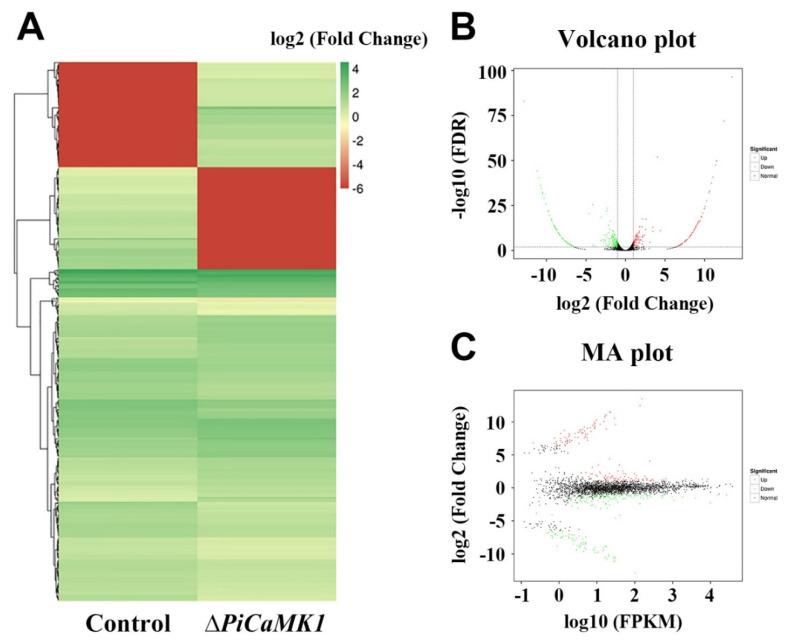
Transcriptome analysis of the differentially expressed genes (DEGs) between the control and Δ*PiCaMK1* strains. (**A**) Clustering (heatmap) analysis of the DEGs. (**B**) Volcano plot analysis of the DEGs. (**C**) MA plot analysis of the DEGs.

**Figure 9 jof-08-00667-f009:**
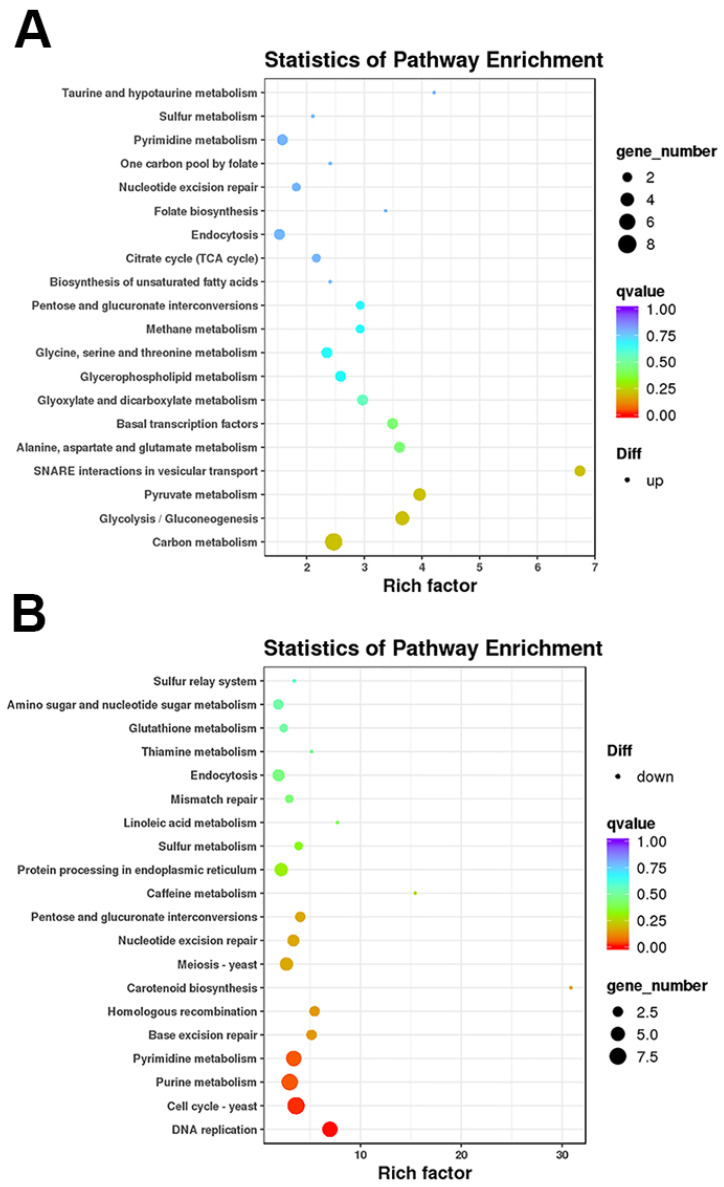
KEGG enrichment of the DEGs between the control and Δ*PiCaMK1* strains. (**A**) Up-regulated DEGs. (**B**) Down-regulated DEGs. Each scatter plot in panel (**A**) or (**B**) shows the top 20 KEGG pathways enriched, and the red color indicates the most significant enrichment.

**Figure 10 jof-08-00667-f010:**
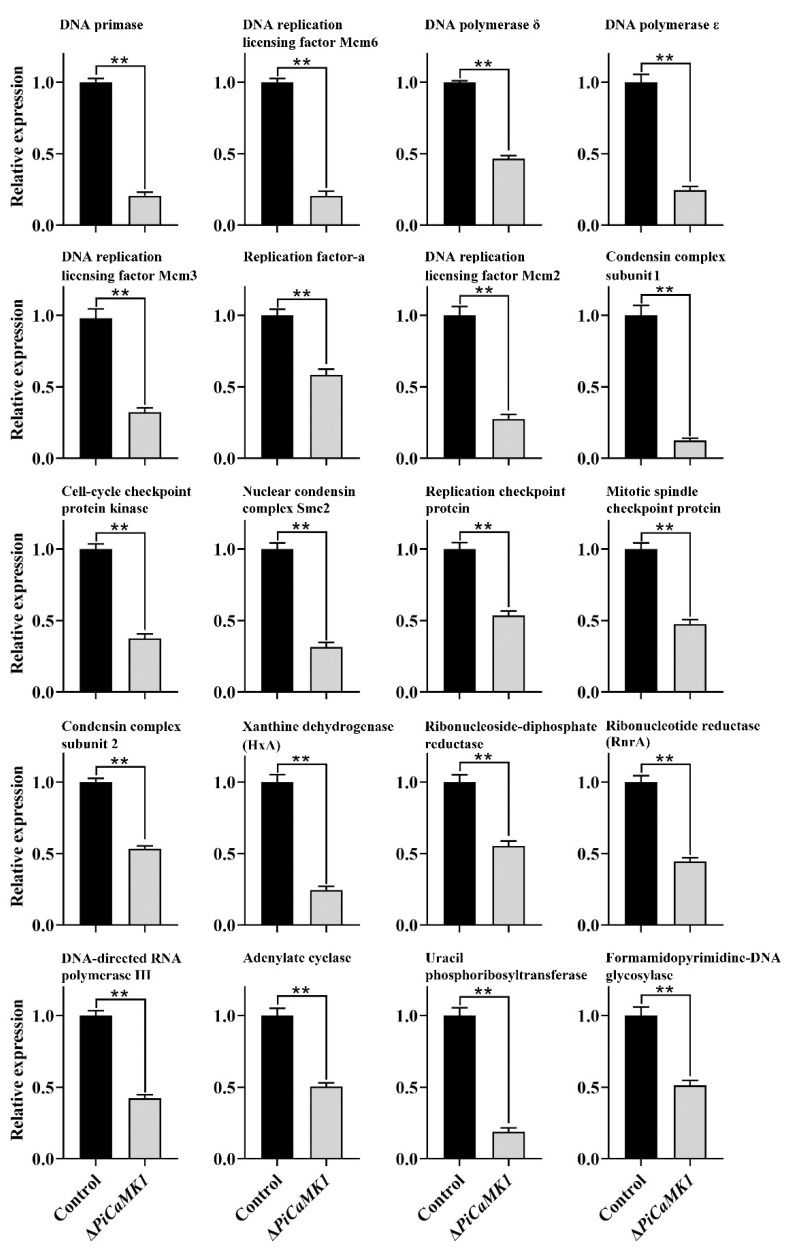
RT-qPCR validation of DEGs identified in the present transcriptome analysis. The experiments were performed in triplicate with three technical repeats, and the results were expressed as relative transcript abundance with mean ± SD. One-way ANOVA and the least significant difference (LSD) test were applied to statistics analysis (** *p* < 0.01).

**Table 1 jof-08-00667-t001:** KEGG-enriched DEGs down-regulated in the *PiCaMK1*-knockout mutant.

KEGG Pathway (ID)	Gene ID	Gene Function	Log_2_ * FC	FDR
DNA replication (ko03030)	EKV13547	DNA primase (large subunit)	−7.57	1.06 × 10^−5^
	EKV11776	DNA replication licensing factor Mcm6	−2.13	3.01 × 10^−7^
	EKV18089	DNA polymerase δ (catalytic subunit)	−1.53	5.95 × 10^−5^
	EKV08814	DNA polymerase ε (catalytic subunit)	−1.49	2.40 × 10^−4^
	EKV11198	DNA replication licensing factor Mcm3	−1.19	3.90 × 10^−5^
	EKV17606	Replication factor-a protein	−1.14	2.22 × 10^−3^
	EKV16128	DNA replication licensing factor Mcm2	−1.08	2.62 × 10^−5^
Cell cycle (ko04111)	EKV11033	Condensin complex subunit (HEAT-like repeat)	−4.11	1.19 × 10^−3^
	EKV11776	DNA replication licensing factor Mcm6	−2.13	3.01 × 10^−7^
	EKV17483	Cell-cycle checkpoint protein kinase (DNA damage response protein kinase)	−2.12	1.27 × 10^−10^
	EKV19093	Nuclear condensin complex Smc2 (structural maintenance of chromosome)	−1.51	7.96 × 10^−8^
	EKV04214	Replication checkpoint protein (MRC1-like domain)	−1.38	1.92 × 10^−3^
	EKV16186	Mitotic spindle checkpoint protein (Mad2)	−1.27	3.21 × 10^−3^
	EKV11198	DNA replication licensing factor Mcm3	−1.19	3.90 × 10^−5^
	EKV16749	Condensin complex subunit 1	−1.10	3.82 × 10^−4^
	EKV16128	DNA replication licensing factor Mcm2	−1.08	2.62 × 10^−5^
Purine metabolism (ko00230)	EKV04683	Xanthine dehydrogenase HxA	−7.57	1.06 × 10^−5^
	EKV13547	DNA primase (large subunit)	−7.57	1.06 × 10^−5^
	EKV18089	DNA polymerase δ (catalytic subunit)	−1.53	5.95 × 10^−5^
	EKV19574	Ribonucleoside-diphosphate reductase	−1.53	2.12 × 10^−5^
	EKV08814	DNA polymerase ε (catalytic subunit)	−1.49	2.40 × 10^−4^
	EKV15599	Ribonucleotide reductase RnrA	−1.13	1.08 × 10^−5^
	EKV16890	DNA-directed RNA polymerase III	−1.12	1.81 × 10^−4^
	EKV07940	Adenylate cyclase	−1.01	2.19 × 10^−4^
Pyrimidine metabolism (ko00240)	EKV07761	Uracil phosphoribosyltransferase	−10.48	1.79 × 10^−32^
	EKV13547	DNA primase (large subunit)	−7.57	1.06 × 10^−5^
	EKV18089	DNA polymerase δ (catalytic subunit)	−1.53	5.95 × 10^−5^
	EKV19574	Ribonucleoside-diphosphate reductase	−1.53	2.12 × 10^−5^
	EKV08814	DNA polymerase ε (catalytic subunit)	−1.49	2.40 × 10^−4^
	EKV15599	Ribonucleotide reductase RnrA	−1.13	1.08 × 10^−5^
	EKV16890	DNA-directed RNA polymerase III	−1.12	1.81 × 10^−4^
Base excision repair (ko03410)	EKV18089	DNA polymerase δ (catalytic subunit)	−1.53	5.95 × 10^−5^
	EKV08814	DNA polymerase ε (catalytic subunit)	−1.49	2.40 × 10^−4^
	EKV07371	Formamidopyrimidine-DNA glycosylase	−1.33	5.95 × 10^−5^
Nucleotide excision repair (ko03420)	EKV18089	DNA polymerase δ (catalytic subunit)	−1.53	5.95 × 10^−5^
	EKV08814	DNA polymerase ε (catalytic subunit)	−1.49	2.40 × 10^−4^
	EKV15299	DNA repair protein RAD1	−1.39	6.80 × 10^−7^
	EKV17606	Replication factor-a protein	−1.14	1.22 × 10^−3^
Mismatch repair (ko03430)	EKV18089	DNA polymerase δ (catalytic subunit)	−1.53	5.95 × 10^−5^
	EKV17606	Replication factor-a protein	−1.14	1.22 × 10^−3^
Meiosis (ko04113)	EKV11776	DNA replication licensing factor Mcm6	−2.13	3.01 × 10^−7^
	EKV16186	Mitotic spindle checkpoint protein (Mad2)	−1.27	3.21 × 10^−3^
	EKV11198	DNA replication licensing factor Mcm3	−1.19	3.90 × 10^−5^
	EKV16128	DNA replication licensing factor Mcm2	−1.08	2.62 × 10^−5^
	EKV07940	Adenylate cyclase	−1.01	2.19 × 10^−4^
MAPK signaling pathway (ko04011)	EKV17484	Phosphatidylinositol 4-kinase	−1.62	3.94 × 10^−6^
Oxidative phosphorylation (ko00190)	EKV05405	Cytochrome c oxidase assembly protein	−7.43	3.13 × 10^−5^
	EKV18906	Mitochondrial F_1_/F_0_-ATP synthase	−4.60	3.19 × 10^−21^
Carotenoid biosynthesis (ko00906)	EKV07272	Aldehyde dehydrogenase (β-apo-4′-carotenal oxygenase)	−7.28	9.59 × 10^−5^
Glutathione metabolism (ko00480)	EKV19574	Ribonucleoside-diphosphate reductase	−1.53	2.12 × 10^−5^
	EKV15599	Ribonucleotide reductase RnrA	−1.13	1.08 × 10^−5^
Cysteine and methionine metabolism (ko00270)	EKV06483	Cysteine synthase A	−9.77	2.13 × 10^−22^
Sulfur metabolism (ko00920)	EKV06483	Cysteine synthase A	−9.77	2.13 × 10^−22^
	EKV18475	Assimilatory sulfite reductase	−1.73	5.56 × 10^−3^
Starch and sucrose metabolism (ko00500)	EKV04855	Oligo-1,6-glucosidase (α-amylase or maltase)	−3.09	3.67 × 10^−6^
Amino sugar and nucleotide sugar metabolism (ko00520)	EKV05685	Glucosamine-6-phosphate deaminase	−12.86	1.32 × 10^−83^
	EKV11299	NADH-cytochrome b_5_ reductase	−1.59	1.52 × 10^−8^
	EKV15950	Chitin synthase A/B	−1.19	5.23 × 10^−4^
Biosynthesis of amino acids (ko01230)	EKV06483	Cysteine synthase A	−9.77	2.13 × 10^−22^
	EKV17406	Catabolic 3-dehydroquinase	−1.26	9.29 × 10^−5^
Protein processing in endoplasmic reticulum (ko04141)	EKV13467	DnaJ-related protein SCJ1	−8.53	1.19 × 10^−10^
	EKV14522	Polyubiquitin binding protein (Doa1/Ufd3)	−1.68	1.34 × 10^−9^
	EKV14057	Heat shock protein 90 (HSP90)	−1.33	8.03 × 10^−7^
	EKV13033	Heat shock 70 kDa protein (HSP70)	−1.09	2.44 × 10^−6^
	EKV13686	Nuclear protein localization protein (NPL4 family)	−1.04	6.08 × 10^−4^
Endocytosis (ko04144)	EKV18650	Phospholipase D	−9.35	1.16 × 10^−17^
	PHI26_NewGene_31	Vacuolar protein sorting-associated protein (VHS domain)	−1.57	4.21 × 10^−8^
	PHI26_NewGene_32	Vacuolar protein sorting-associated protein (FYVE-like protein)	−1.10	5.12 × 10^−3^
	EKV13033	Heat shock 70 kDa protein (HSP70)	−1.09	2.44 × 10^−6^

* FC = Fold Change; FDR, False Discovery Rate (i.e., the corrected *p*-value also named *q*-value).

## Data Availability

Not applicable.

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
