# Peer review of "Characterization and Functional Analysis of a New Calcium/Calmodulin-Dependent Protein Kinase (CaMK1) in the Citrus Pathogenic Fungus Penicillium italicum"

_jof, 2022, doi:10.3390/jof8070667_

Round 1

Reviewer 1 Report

I have no major comments on the manuscript submitted for review. It is of a good scientific and technical level. The individual chapters are well described and do not raise any reservations. Therefore, in my opinion, the work will be a good supplement to the current knowledge about the research undertaken. A slight remark, please use cursive to write the names of the fungi in Fig. 2.

Author Response

Truly thank you for your comment. In the current revision, we have re-edited the names of the fungi in Figure 2 in cursive style to make them clearer. The manuscript with the revised Figure 2 is attached with this letter.

Reviewer 2 Report

In general the manuscript is relevant for its publication

i have some minor concerns 

In figure 3 panel B  it would be much better for the readers to include the amplicon size with the oligonucleotides used for genetic analysis and also the expected bands in the southern blot

Figure 5. virulence in a fruit?

the fruit is already separated from the plant, I am agree with the damage of the postharvest fungal pathogen in the fruit, but virulence? I am not agree with this terminology because the fruit is already separated from the plant, and is weird to expect that the protection systems of the plant-fruit will be in optimal conditions. 

Author Response

Truly thank you for your comments. In the current revision, we have added the probe size information (310 bp) in Figure 3 panel A as well as in the Figure 3 legend (line 276). We agree with your suggestion that the term ‘citrus fruit’ used to illustrate Figure 5 is not accurate, so we add ‘postharvest’ before the ‘citrus fruit’ and its alternative terms, please see at Figure 5 legend (lines 310-311) and at lines 168, 297, 301, and 305. Please note, all the revisions are highlighted with yellow color in the attached manuscript.

This manuscript is a resubmission of an earlier submission. The following is a list of the peer review reports and author responses from that submission.